# Global trends and disparities in ischemic heart disease attributable to high systolic blood pressure, 1990–2021: Insights from the global burden of disease study

Zhenhua Zhuang[1], Qiuju Wang[2], Haifeng Li[3], Shenghong Lan[1], Yanru Su[1], Yue Lin[1], Peng Guo[1]*

1 Department of Cardiovascular Medicine, Guigang City People's Hospital, Guigang, Guangxi, China, 2 Health Management Center, Guigang City People's Hospital, Guigang, Guangxi, China, 3 Department of Anesthesiology and Surgery, Shenzhen Pingle Orthopedic Hospital, Shenzhen, Guangdong, China

* gpgxx8290@sina.com

## Abstract

### Objective

To comprehensively assess the global, regional, and national burden of ischemic heart disease (IHD) attributable to high systolic blood pressure (HSBP).

### Methods

Using the Global Burden of Disease (GBD) 2021 dataset, we conducted a systematic analysis of mortality, disability-adjusted life years (DALYs), and their age-standardized rates (ASRs) for HSBP-related IHD from 1990 to 2021. We stratified data by sex, age (25–95+ years), and sociodemographic index (SDI) categories (Low, Low-Middle, Middle, High-Middle, and High), and examined geographic disparities across 21 regions and 204 countries. Temporal trends were assessed using estimated annual percentage change (EAPC), and smoothed curve modeling and Spearman's correlation were applied to evaluate associations between SDI and ASRs.

### Results

Globally, the absolute number of DALYs attributable to HSBP-related IHD rose substantially from 1990 to 2021, although ASRs decreased over the same period. Males consistently shouldered a larger proportion of the burden, yet women experienced a relatively faster decline in DALY and mortality ASRs. Notable regional disparities were observed, with Central Asia, Eastern Europe, and North Africa & Middle East demonstrating higher burdens despite downward trends, whereas certain low-to-middle SDI regions and select Asian and African countries exhibited rising ASRs. Variations in SDI also correlated with shifting patterns in HSBP-related IHD burden, highlighting the importance of socioeconomic factors.

**Data availability statement:** All data are available without restriction from the Global Health Data Exchange query tool (http://ghdx. healthdata.org/gbd-results-tool).

**Funding:** The author(s) received no specific funding for this work.

**Competing interests:** The authors have declared that no competing interests exist.

## Conclusion

This study underscores the significant global burden of IHD attributable to HSBP, with substantial heterogeneity by sex, age, and geographic setting. The findings emphasize the need for context-specific public health interventions, such as intensified hypertension screening, early detection, and management strategies.

## Introduction

Ischemic heart disease (IHD) represents a significant global health challenge, being one of the leading causes of mortality and morbidity across various populations[1–3]. The burden of IHD extends beyond individual health, imposing substantial economic strains on healthcare systems and societies alike, as evidenced by millions of deaths and considerable disability-adjusted life years (DALYs) lost annually[4]. Previous studies underscores that IHD arises from a complex interplay of factors, including atherosclerotic progression, elevated blood pressure, and behavioral risks such as sedentary lifestyles and unbalanced dietary patterns [5–7]. These factors collectively drive coronary artery narrowing and reduced myocardial perfusion, ultimately leading to acute events such as myocardial infarction.

Several well-established modifiable risk factors contribute to IHD, including dyslipidemia, obesity, diabetes, smoking, and poor dietary habits. High systolic blood pressure (HSBP) is one of the most significant modifiable risk factors for IHD [8,9]. Chronic HSBP contributes to vascular endothelial dysfunction, arterial stiffening, and myocardial oxygen imbalance, accelerating atherosclerosis and coronary artery disease progression[10]. Long-term exposure to elevated blood pressure promotes left ventricular hypertrophy, myocardial fibrosis, and microvascular dysfunction, reducing coronary perfusion and increasing the likelihood of acute coronary events[11,12]. Furthermore, oxidative stress, inflammation, and thrombogenesis associated with hypertension amplify cardiovascular risk[13,14].

While dyslipidemia, diabetes, and hypertension are independently associated with IHD, they often interact synergistically rather than act in isolation. Chronic HSBP exacerbates metabolic disturbances such as insulin resistance and dyslipidemia, further compounding cardiovascular risk. Conversely, diabetes and dyslipidemia contribute to endothelial dysfunction and arterial stiffness, amplifying hypertension-related vascular damage. Lifestyle factors such as smoking and an unhealthy diet also accelerate atherosclerosis, with their effects being particularly pronounced in hypertensive individuals[15,16]. Given this complex interplay, controlling HSBP remains a cornerstone of cardiovascular disease prevention, particularly in individuals with multiple coexisting risk factors.

Recognizing this complexity, the 2024 ESC Hypertension Guidelines introduced the concept of "elevated blood pressure" (120–139 mmHg systolic), underscoring the continuous nature of cardiovascular risk and the importance of early intervention[17]. Notably, clinical trials have demonstrated that targeted blood pressure reduction in individuals with elevated BP, particularly those with additional cardiovascular risk

factors, significantly lowers the incidence of future cardiovascular events[18,19]. These findings challenge conventional treatment paradigms and emphasize the need for proactive blood pressure management even among individuals previously classified as normotensive. Although HSBP is a well-established contributor to ischemic heart disease, it remains a "silent burden" in clinical practice—frequently under-recognized in cardiovascular risk stratification and undertreated in primary prevention. Recent findings by Cesaro et al. highlight a persistent disconnect between physician-estimated and calculated cardiovascular risk, indicating that elevated blood pressure is often undervalued in clinical decision-making, potentially delaying timely intervention [20].

Despite extensive research on hypertension and IHD, a comprehensive global epidemiological assessment quantifying the burden of IHD attributable to HSBP remains limited. To address this gap, this study leverages the Global Burden of Disease (GBD) 2021 dataset, which provides standardized estimates of mortality, prevalence, and DALYs associated with HSBP and IHD across 204 countries from 1990 to 2021. By analyzing temporal trends, sex- and age-specific variations, and socioeconomic disparities, this study aims to provide actionable insights for guiding global cardiovascular prevention strategies.

## Materials and methods

### Data source

The 2021 GBD study leveraged the latest epidemiological data and refined, standardized methods to comprehensively evaluate health losses associated with 371 diseases, injuries, and conditions, as well as 88 risk factors, across 204 countries and territories[21,22]. The cause-of-death database comprises seven types of data sources: vital registration, verbal autopsy, cancer registries, police records, sibling history, surveillance, surveys/censuses, and minimally invasive tissue sampling diagnostics. Statistical methods are applied to enhance the comparability of mortality data sources, including reclassification of nonspecific or unspecified codes, noise reduction algorithms, and the Cause of Death Ensemble model (CODem). DisMod-MR 2.1, a Bayesian disease modeling meta-regression tool, generates internally consistent mortality estimates stratified by sex, location, year, and age group. Additionally, this tool estimates cascading prevalence across five levels of the GBD geographic hierarchy to derive epidemiological indicators for locations lacking primary epidemiological data. Epidemiological data from higher levels of the hierarchy serve as prior information for estimating epidemiological parameters at lower levels.

Disability-adjusted life years (DALYs) are the sum of years lived with disability (YLDs) and years of life lost (YLLs). YLDs are calculated through a microsimulation process that incorporates the estimated prevalence of nonfatal sequelae (the consequences of diseases or injuries) by age, sex, location, and year, along with the disability weights assigned to each sequela. YLLs are computed as the product of the estimated number of deaths by age, sex, location, and year and the standard expected life expectancy at the age of death for a given cause.

Population attributable fraction (PAF) is calculated to quantify the contribution of HSBP to the burden of IHD. The burden of mortality or DALYs attributable to HSBP can be estimated by multiplying the location-, year-, age-, and sex-specific PAF by the corresponding DALYs or mortality for IHD. Detailed information on the study design and methods, including the calculation of risk factor attribution, is extensively described in existing GBD literature.

### Definitions

GBD 2021 defines HSBP as ≥115 mmHg based on the theoretical minimum risk exposure level (TMREL), which represents the exposure level associated with the lowest population risk level [23]. TMREL represents the exposure level at which the likelihood of developing any HSBP-related disease is minimized. To determine the TMREL for HSBP, the Global Burden of Disease (GBD) collaborators analyzed systolic blood pressure values reported in randomized controlled trials and prospective cohort studies. In each study, they identified the reference group as the cohort with the lowest SBP values. The lower bound of the TMREL was set as the 15th percentile of the lower limit of the reference range across all

studies, while the upper bound was defined as the 15th percentile of the midpoint between the lower and upper limits of the reference group's SBP. This approach resulted in a new TMREL range of 105–115 mmHg, accounting for the uncertainty in defining the precise threshold for minimal risk.

IHD was categorized as a Level 3 cause under "Non-communicable diseases" and "Cardiovascular diseases." To identify IHD cases, the study applied International Classification of Diseases, Ninth Revision (ICD-9) codes 410–414.9 and Tenth Revision (ICD-10) codes I20–I25.9, covering acute myocardial infarction, chronic IHD, and chronic stable angina according to the Fourth Universal Definition of Myocardial Infarction and the Rose Angina Questionnaire[24].

## Data collection

We accessed and retrieved global burden data on IHD attributable to HSBP from 1990 to 2021 using the Global Health Data Exchange (GHDx) query tool (http://ghdx.healthdata.org/gbd-results-tool). The data were sourced from 21 regions, comprising countries and territories that are geographically proximate and share similar epidemiological characteristics, covering a total of 204 countries and territories. The study variables included DALYs and deaths, with metrics reported as absolute counts, crude rates, and ASRs. To adjust for changes arising from population growth and aging, we used age-standardized rates (per 100,000) for comparisons and trend analyses[25].

$$ASR = \frac{\sum_{i=1}^{A} a_i w_i}{\sum_{i=1}^{A} w_i}$$

where $a_i$ is the age specific rate and $w_i$ is the weight in the same age subgroup of the chosen reference standard population (in which i denotes the i^"th" age class) and A is the upper age limit.

We grouped the data by sex, age, and five SDI categories—Low, Low-Middle, Middle, High-Middle, and High—to systematically estimate the burden. The SDI categories were assigned at the country or territory level. SDI is a country-level composite measure incorporating educational attainment (mean years of schooling), income per capita (log-transformed), and fertility rate (total fertility rate among women under 25 years old). Since the GBD 2021 dataset does not include estimates for the burden of IHD attributable to HSBP among individuals younger than 25 years, our analysis primarily focused on populations aged 25 years and older, with age groups categorized in five-year intervals from 25 to 95+years. All data were reported with 95% uncertainty intervals (UI). Since this study is based on publicly available data, no ethical approval was required.

## Statistical analysis

We used the estimated annual percentage change (EAPC) in ASRs to assess temporal trends. Assuming a linear relationship between the natural logarithm of ASR and time, we applied the model: $Y = \alpha + \beta X + \varepsilon$, where Y represents $\ln(ASR)$, X denotes calendar year, $\varepsilon$ is the error term, and $\beta$ is the regression coefficient. EAPC was computed using the formula $EAPC_{with\ 95\%CI} = 100 \times (\exp(\beta) - 1)$ [26]. If the EAPC and its lower confidence bound are both positive, the ASRs exhibit an increasing trend; conversely, if the EAPC and its upper confidence bound are both negative, the ASRs demonstrate a decreasing trend. Additionally, we employed smoothed curve modeling and Spearman's correlation coefficients to investigate the association between SDI and ASRs. The strength of correlation was determined based on Spearman correlation coefficient (r) values (r = 0.2–0.39, weak correlation; r = 0.4–0.59, moderate correlation; r = 0.6–0.79, strong correlation; r > 0.8, very strong correlation) [27]. All statistical analyses were performed using R (version 4.3.1), with p < 0.05 considered statistically significant. The R codes are recorded in S1 File.

## Results

### Global burden

Globally, the absolute number of DALYs increased from 58,868,198 (95% UI: 47,917,170–68,473,028) in 1990–93,959,103 (95% UI: 75,572,860–110,787,955) in 2021, marking an approximate 60% rise. Males accounted for around

60% of these DALYs, while females made up about 40%. Over the same period, total deaths rose from 2,851,404 (95% UI: 2,318,619–3,301,026) in 1990–4,692,863 (95% UI: 3,799,886–5,515,763) in 2021, an increase of roughly 39%. Of these deaths, approximately 54% occurred in males and 46% in females (Table 1).

In contrast, age-standardized DALY rates (ASRs) and age-standardized death rates (ASDRs) for IHD attributable to HSBP declined over this interval, as indicated by EAPC values of –1.28 (95% CI: –1.34, –1.23) and –1.43 (95% CI: –1.47, –1.39), respectively. Specifically, the ASR for DALYs decreased from 1,577.61 (95% UI: 1,289.46–1,832.77) to 1,101.20 (95% UI: 886.46–1,299.09), and ASDR from 85.74 (95% UI: 68.85–99.44) to 56.73 (95% UI: 45.78–66.69) (Table 1). Between 1990 and 2021, males consistently showed a higher IHD burden attributable to HSBP compared to females. However, the relative decline in female DALY and mortality ASRs (–35% and –38%) exceeded that in males (–30% and –34%), as shown in Fig 1.

### Sex- and age-specific burden

The burden of HSBP-related IHD for both males and females was predominantly concentrated in older age groups (Fig 2 and S1 Table). Complete DALYs and death rates by sex and age group are reported in S1 Table. In nearly all age categories, males experienced higher DALYs and mortality rates than females (Fig 2). The DALY and mortality rates generally rose with advancing age for both sexes. After age 95, rates started to decline among males, whereas the burden in females continued to grow (Fig 2).

### Regional and national differences

Table 1 and Fig 3 illustrate the burden and EAPC across 21 regions. In 2021, Central Asia had the highest age-standardized DALY [2,699.80 (95% UI: 2,201.59–3,195.76)] and ASDR [150.54 (95% UI: 122.75–177.20)], although both exhibited declining trends from 1990 to 2021 [EAPC = –1.11 (95% CI: –1.43, –0.78) and –0.82 (95% CI: –1.10, –0.54)]. Eastern Europe and North Africa & Middle East also had high HSBP-related IHD burdens, following a similar downward trajectory (Table 1). In contrast, High-income Asia Pacific reported the lowest burden, with an age-standardized DALY of 231.76 (95% UI: 178.68–278.71) and an ASDR of 12.44 (95% UI: 9.30–15.17). Australasia and Western Europe ranked second and third lowest in terms of ASRs. Of note, between 1990 and 2021, only six regions—East Asia, Western Sub-Saharan Africa, Eastern Sub-Saharan Africa, Oceania, South Asia, and Southeast Asia—showed increasing burdens. East Asia experienced the steepest rises [EAPC = 1.19 (95% CI: 0.87, 1.52) and 1.50 (95% CI: 1.14, 1.87)].

A comparable pattern emerged at the national level (Fig 4 and S2 Table). In 2021, among 204 countries, Nauru reported especially high burdens, with age-standardized DALYs of 5,803.73 (95% UI: 4,125.07–7,778.20) and an ASDR of 236.42 (95% UI: 172.35–303.37). Vanuatu, Ukraine, and Belarus followed closely behind. Conversely, the Republic of Korea had the lowest age-standardized DALYs [181.98 (95% UI: 126.35–235.22)] and ASDR [236.42 (95% UI: 172.35–303.37)], alongside San Marino and Japan, which also presented minimal burdens (Fig 4). Denmark, the United Kingdom, and Israel exhibited the lowest EAPC values, indicating the fastest decreases, whereas several African nations—Lesotho, Zimbabwe, and Cameroon—had the highest EAPCs (Fig 4 and S2 Table).

### Age-specific trends across regions

At the global level, the EAPC of age-standardized DALYs and ASDR showed an overall decline, particularly in individuals aged 45 years and older, indicating the effectiveness of long-term hypertension management efforts in middle-aged and elderly populations. However, distinct regional variations exist in age-specific trends. Australasia, Western Europe, High-income Asia Pacific, and High-income North America showed a consistent decline in EAPC values across all age groups, particularly in the elderly population (≥45 years), suggesting effective hypertension screening, treatment accessibility, and preventive health measures in these high-income regions. In contrast, East Asia, South Asia, Andean Latin America, Eastern Sub-Saharan Africa, Western Sub-Saharan Africa, and Oceania demonstrated positive EAPC values in

**Table 1. The DALYs, deaths, and their estimated annual percentage changes of ischemic heart disease attributable to high systolic blood pressure from 1990 to 2021.**

| Location | DALYs | | | | |
|---|---|---|---|---|---|
| | Number of cases, 1990 | ASR, 1990 | Number of cases, 2021 | ASR, 2021 | |
| Global | 58868198 (47917170,68473028) | 1577.61 (1289.46,1832.77) | 93959103 (75572860,110787955) | 1101.2 (886.46,1299.09) | |
| **SDI** | | | | | |
| Low SDI | 2623115 (2005409,3190344) | 1188.71 (913.41,1448.65) | 5774073 (4492360,7003434) | 1168.22 (908.24,1408.71) | |
| Low-middle SDI | 8633649 (6792966,10407345) | 1429.86 (1139.3,1716.96) | 22325108 (17780577,26537204) | 1561.5 (1254.1,1851.49) | |
| Middle SDI | 11424455 (8914487,13660149) | 1174.39 (921.72,1395.08) | 30264857 (24155348,36087917) | 1165.32 (934.33,1386.69) | |
| High-middle SDI | 17922354 (14678810,20718149) | 1932.89 (1588.69,2238.44) | 24156099 (19693562,28540578) | 1233.71 (1003.23,1457.33) | |
| High SDI | 18161300 (14988343,20856135) | 1640.86 (1349.84,1887.37) | 11340819 (8937914,13525194) | 529.32 (419.56,631.48) | |
| **Region** | | | | | |
| Andean Latin America | 105881 (73070,138141) | 544.5 (377.63,704.89) | 277885 (199750,354529) | 478.62 (344.7,608.7) | |
| Australasia | 429498 (350926,496068) | 1837.29 (1495.47,2125.52) | 213516 (166521,258829) | 383.24 (301,463.49) | |
| Caribbean | 425626 (338918,512570) | 1691.55 (1351.9,2027.27) | 624787 (470345,778686) | 1156.36 (870.32,1440.86) | |
| Central Asia | 1456930 (1185640,1700484) | 3262.48 (2653.79,3809.63) | 2013486 (1630900,2399947) | 2699.8 (2201.59,3195.76) | |
| Central Europe | 4436580 (3712709,5047684) | 3081 (2584.21,3508.6) | 3182646 (2595539,3718039) | 1407.69 (1143.95,1647.3) | |
| Central Latin America | 920411 (732286,1092134) | 1165.94 (933.75,1375.36) | 2498724 (1968090,3001015) | 1014.42 (801.87,1215.54) | |
| Central Sub-Saharan Africa | 320983 (227160,427317) | 1529.76 (1101.76,1986.11) | 635263 (458341,844131) | 1249.44 (911.32,1639.37) | |
| East Asia | 5313377 (3802070,6712748) | 713.58 (529.48,899.04) | 18429960 (13963924,23757462) | 910.19 (688.08,1155.36) | |
| Eastern Europe | 9074898 (7516388,10374817) | 3412.08 (2822.97,3912.17) | 9413408 (7573370,11040648) | 2672.56 (2140.9,3138.44) | |
| Eastern Sub-Saharan Africa | 425117 (310573,534676) | 587.7 (433.8,735.21) | 1233990 (917600,1518073) | 740.45 (561.14,907.37) | |
| High-income Asia Pacific | 1257056 (1033862,1454698) | 658.72 (535.99,762.37) | 1121685 (847687,1357777) | 231.76 (178.68,278.71) | |
| High-income North America | 6250381 (5060133,7278861) | 1757.64 (1421.61,2046.91) | 4169531 (3128478,5080505) | 628.67 (474.21,769.96) | |
| North Africa and Middle East | 4439807 (3456761,5314599) | 2745.95 (2166.91,3266.4) | 8998133 (6934974,11059250) | 2051.54 (1604.08,2501.8) | |
| Oceania | 42274 (29434,56116) | 1415.5 (1002.68,1855.8) | 127468 (89111,168098) | 1604.04 (1133.86,2087.95) | |
| South Asia | 8432743 (6584397,10210683) | 1432.2 (1130.16,1718) | 23433547 (18649640,28410721) | 1587.16 (1269.85,1917.32) | |
| Southeast Asia | 3084463 (2370751,3707827) | 1232.75 (957.01,1478.67) | 8299217 (6626609,9964769) | 1282.03 (1030.82,1534.66) | |
| Southern Latin America | 542123 (415248,656393) | 1226.74 (947.3,1480.29) | 480929 (383076,567933) | 546.45 (435.22,645.52) | |
| Southern Sub-Saharan Africa | 241162 (188680,290018) | 897.27 (702.46,1075.38) | 531111 (414772,624299) | 957.47 (754.98,1121.91) | |
| Tropical Latin America | 1281644 (1020927,1511657) | 1428.95 (1144.38,1685.24) | 1896937 (1530974,2236948) | 733.29 (592.06,864.03) | |
| Western Europe | 9585759 (7979160,10910222) | 1638.5 (1359.21,1870.03) | 4313795 (3363492,5126721) | 432.71 (341.66,510.65) | |
| Western Sub-Saharan Africa | 801484 (607108,992114) | 982.62 (748.36,1207.44) | 2063084 (1625179,2550114) | 1148.22 (913.55,1397.95) | |

DALYs, disability-adjusted life years; ASR, age-standardized rate; EAPC, estimated annual percentage change.

| | Deaths | | | | |
|---|---|---|---|---|---|
| EAPC | Number of cases, 1990 | ASR, 1990 | Number of cases, 2021 | ASR, 2021 | EAPC |
| -1.28(-1.34,-1.23) | 2851404 (2318619,3301026) | 85.74 (68.85,99.44) | 4692863 (3799886,5515763) | 56.73 (45.78,66.69) | -1.43(-1.47,-1.39) |
| | | | | | |
| -0.01(-0.09,0.06) | 103505 (79572,125443) | 55.87 (43.57,67.64) | 236385 (183760,285823) | 56.97 (44.48,68.64) | 0.19(0.09,0.29) |
| 0.42(0.36,0.48) | 343573 (274516,412431) | 66.88 (53.57,80.26) | 942428 (761327,1111835) | 73.98 (60,87.19) | 0.49(0.41,0.56) |
| 0.04(-0.03,0.12) | 482769 (380774,572686) | 60.36 (47.46,71.61) | 1455322 (1169452,1723445) | 61.55 (49.32,72.64) | 0.2(0.11,0.3) |
| -1.77(-2.04,-1.5) | 898492 (738716,1036325) | 108.78 (88.31,125.75) | 1356637 (1101517,1600031) | 70.22 (56.73,83.01) | -1.64(-1.85,-1.42) |
| -3.99(-4.18,-3.8) | 1017937 (826854,1169595) | 92.01 (74.43,105.93) | 696916 (538083,838241) | 28.8 (22.42,34.55) | -4.09(-4.25,-3.93) |
| | | | | | |
| -0.33(-0.79,0.12) | 5232 (3634,6639) | 29.29 (20.55,37.38) | 14939 (10901,18961) | 26.67 (19.5,33.77) | -0.16(-0.6,0.28) |
| -5.41(-5.61,-5.21) | 23478 (18869,27141) | 102.7 (82.09,119.3) | 13769 (10295,16906) | 22.41 (16.95,27.34) | -5.26(-5.42,-5.1) |
| -1.21(-1.4,-1.02) | 20875 (16678,24976) | 88.7 (70.89,106.1) | 30240 (23037,37046) | 55.39 (42.07,67.89) | -1.52(-1.71,-1.33) |
| -1.11(-1.43,-0.78) | 70707 (57651,82449) | 172.08 (139.2,201.17) | 98830 (80215,116479) | 150.54 (122.75,177.2) | -0.82(-1.1,-0.54) |
| -2.87(-2.99,-2.76) | 223230 (188845,253069) | 166.22 (139.65,189.04) | 191126 (153700,223294) | 80.37 (64.68,93.97) | -2.65(-2.76,-2.54) |
| -0.6(-0.82,-0.37) | 43107 (34432,50781) | 61.91 (49.65,72.8) | 129358 (102777,154267) | 54.63 (43.42,65.11) | -0.49(-0.7,-0.27) |
| -0.9(-0.98,-0.81) | 12720 (9105,16665) | 74.95 (54.54,96.64) | 25592 (18658,33653) | 62.8 (45.57,82.55) | -0.81(-0.9,-0.73) |
| 1.19(0.87,1.52) | 238173 (175715,299279) | 40.93 (30.07,50.91) | 1041949 (800878,1322946) | 56.1 (42.66,71.82) | 1.5(1.14,1.87) |
| -1.33(-1.84,-0.82) | 460174 (381671,530320) | 187.21 (155.15,215.92) | 526532 (428419,620118) | 146.89 (119.42,173.19) | -1.21(-1.66,-0.77) |
| 0.61(0.47,0.75) | 16752 (12417,20965) | 27.71 (20.63,34.71) | 48746 (36892,59980) | 35.37 (27.18,43.84) | 0.67(0.53,0.8) |
| -3.54(-3.67,-3.41) | 69161 (55530,79947) | 39.21 (31.22,45.39) | 74267 (53807,91892) | 12.44 (9.3,15.17) | -3.81(-3.99,-3.63) |
| -3.83(-4.12,-3.54) | 361928 (287786,423896) | 98.82 (78.39,115.78) | 252947 (188479,310515) | 35.28 (26.21,43.22) | -3.86(-4.1,-3.61) |
| -1.02(-1.05,-0.98) | 187582 (149086,222751) | 137.36 (109.4,162.39) | 399749 (315296,485340) | 106.4 (84.13,127.84) | -0.88(-0.91,-0.84) |
| 0.51(0.37,0.65) | 1520 (1078,1992) | 64.26 (46,83.65) | 4544 (3221,5919) | 70.97 (51.7,91.28) | 0.37(0.26,0.48) |
| 0.42(0.33,0.5) | 315976 (249774,377846) | 63.42 (49.53,76.13) | 976065 (785667,1178788) | 74.13 (59.47,89.17) | 0.65(0.52,0.77) |
| 0.22(0.14,0.3) | 125473 (98078,150553) | 59.28 (46.63,70.67) | 353373 (286965,420007) | 62.18 (50.95,73.62) | 0.23(0.13,0.34) |
| -2.17(-2.31,-2.03) | 29197 (22676,35162) | 70.69 (54.79,85.22) | 26768 (21108,31478) | 29.46 (23.2,34.63) | -2.34(-2.5,-2.18) |
| 0.18(-0.24,0.61) | 10056 (7863,12080) | 43.32 (33.64,52.27) | 23001 (18280,26914) | 48.64 (38.98,57.09) | 0.33(-0.09,0.75) |
| -2.09(-2.14,-2.05) | 54134 (43630,63711) | 69.29 (55.57,81.7) | 83458 (66366,98205) | 33.19 (26.31,39.11) | -2.24(-2.31,-2.17) |
| -4.62(-4.78,-4.46) | 545808 (450750,623152) | 90.93 (74.6,104.16) | 285773 (213058,342319) | 24.76 (18.8,29.59) | -4.52(-4.66,-4.37) |
| 0.67(0.49,0.85) | 36123 (27577,44267) | 51.2 (39.2,62.66) | 91836 (73200,111455) | 61.42 (49.16,73.81) | 0.77(0.6,0.94) |

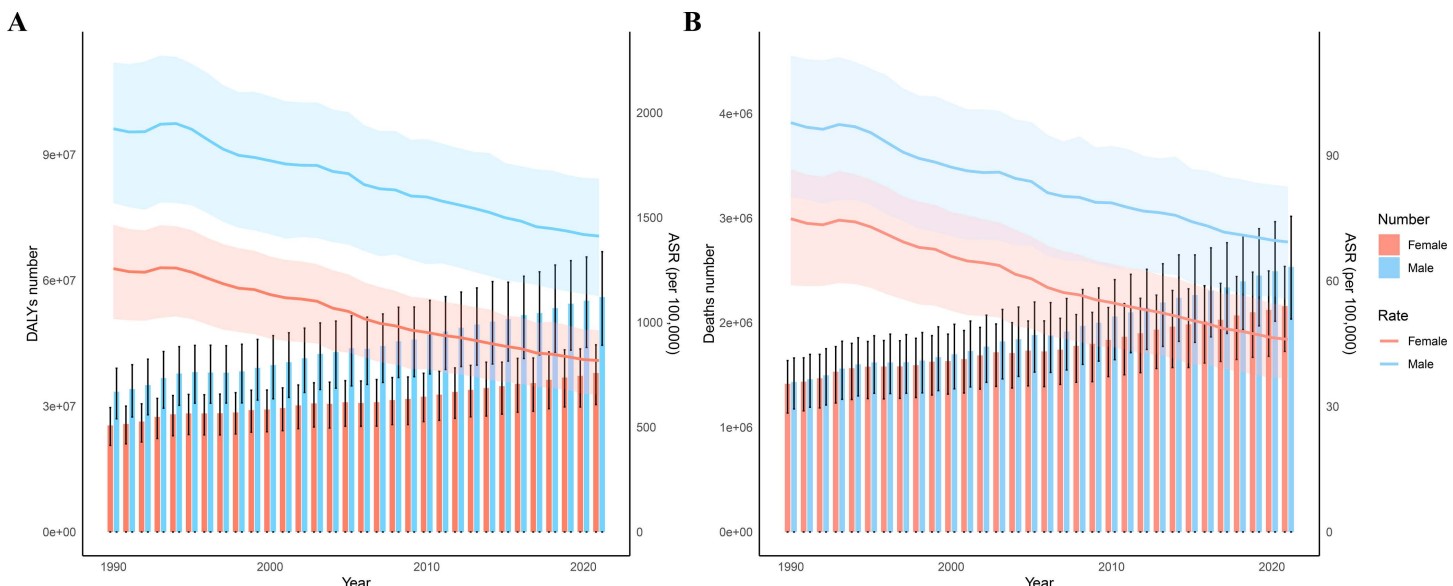

**Fig 1. Global burden of IHD attributable to HSBP from 1990 to 2021.** (A) Numbers and age-standardized rates (per 100,000) of DALYs for IHD attributable to HSBP. (B) Numbers and ASRs of deaths for IHD attributable to HSBP. Error bars and shaded regions indicate 95% uncertainty intervals. DALYs: Disability-adjusted life years; ASR: Age-standardized rate.

the 25–44-year-old age group, indicating a rising burden of HSBP-related IHD among younger populations. Additionally, East Asia, South Asia, and Western Sub-Saharan Africa exhibited a significant increase in IHD burden among individuals aged 75 years and older (Fig 5).

## SDI-based regional and national burden

Among the five SDI levels, Low-Middle SDI regions had the highest age-standardized DALYs [1,561.50 (95% UI: 1,254.10–1,851.49)] and ASDR [73.98 (95% UI: 60.00–87.19)], with the highest EAPC values [0.42 (95% CI: 0.36–0.48) and 0.49 (95% CI: 0.41–0.56)], indicating the fastest growth. Meanwhile, High SDI regions had the lowest age-standardized DALYs [529.32 (95% UI: 419.56–631.48)] and ASDR [28.80 (95% UI: 22.42–34.55)], along with the most pronounced declines [EAPC = −3.99 (95% CI: −4.18, −3.80) and −4.09 (95% CI: −4.25, −3.93)], as shown in Table 1.

Spearman correlation analysis revealed no significant linear association between SDI and IHD burden across the 21 regions (p < 0.05), suggesting that the relationship between socioeconomic development and HSBP-related IHD burden may be nonlinear or influenced by unmeasured confounders. However, among 204 countries and territories, age-standardized DALYs (r = −0.308, p < 0.001) and ASDR (r = −0.262, p < 0.001) exhibited weak negative correlations with SDI, indicating that countries with higher SDI levels tend to have lower IHD burden attributable to HSBP. Given that correlation coefficients between −0.1 and −0.3 typically suggest weak to moderate associations, this finding implies that while SDI is a relevant factor, other determinants such as healthcare policies, population aging, and lifestyle factors likely play a significant role in shaping IHD burden across countries. Further examination via smoothed curve modeling indicated an approximate parabolic relationship between SDI and ASRs from 1990 to 2021. ASRs gradually rose with increasing SDI until approximately SDI ≥ 0.7, after which they declined more sharply. Specifically, SDI values in medium-to-high SDI regions have steadily increased over time, accompanied by decreasing ASRs, whereas low-to-medium SDI regions exhibited both rising SDI and escalating ASRs (Fig 6). Additional analyses showed a similar parabolic relationship between SDI and ASRs across 204 countries and territories, with ASRs climbing in tandem with SDI until about 0.7, followed by a sustained decrease (Fig 6).

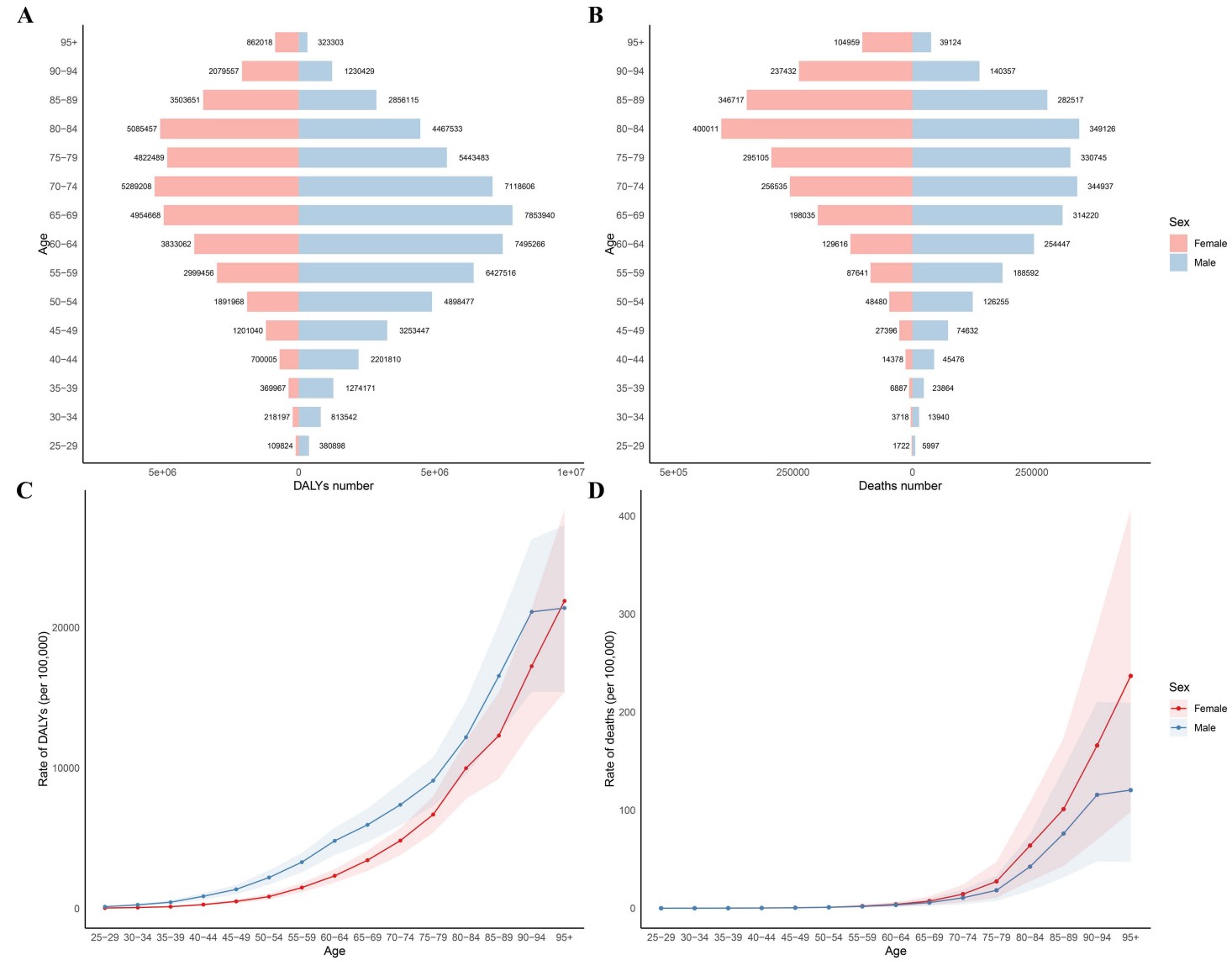

**Fig 2. Age-specific burden of IHD attributable to HSBP for males and females in 2021.** DALYs for males and females across different age groups in 2021(A and C). Mortality rates for males and females across different age groups in 2021(B and D). DALYs: Disability-adjusted life years.

## Discussion

This study offers a comprehensive assessment of the global burden of IHD attributable to HSBP, based on data from the GBD 2021 study across 204 countries and territories from 1990 to 2021. Our findings yield three critical insights with substantial public health implications: (1) Despite a global decline in ASRs of HSBP-related IHD, the absolute burden continues to increase—particularly among men and older adults; (2) In several regions, including East Asia, South Asia, and Sub-Saharan Africa, younger adults (ages 25–44) are experiencing a rising burden, indicating a troubling epidemiologic shift toward earlier onset; (3) Marked regional and socioeconomic disparities persist, with many low- and middle-SDI settings showing rising ASRs and poor hypertension control.

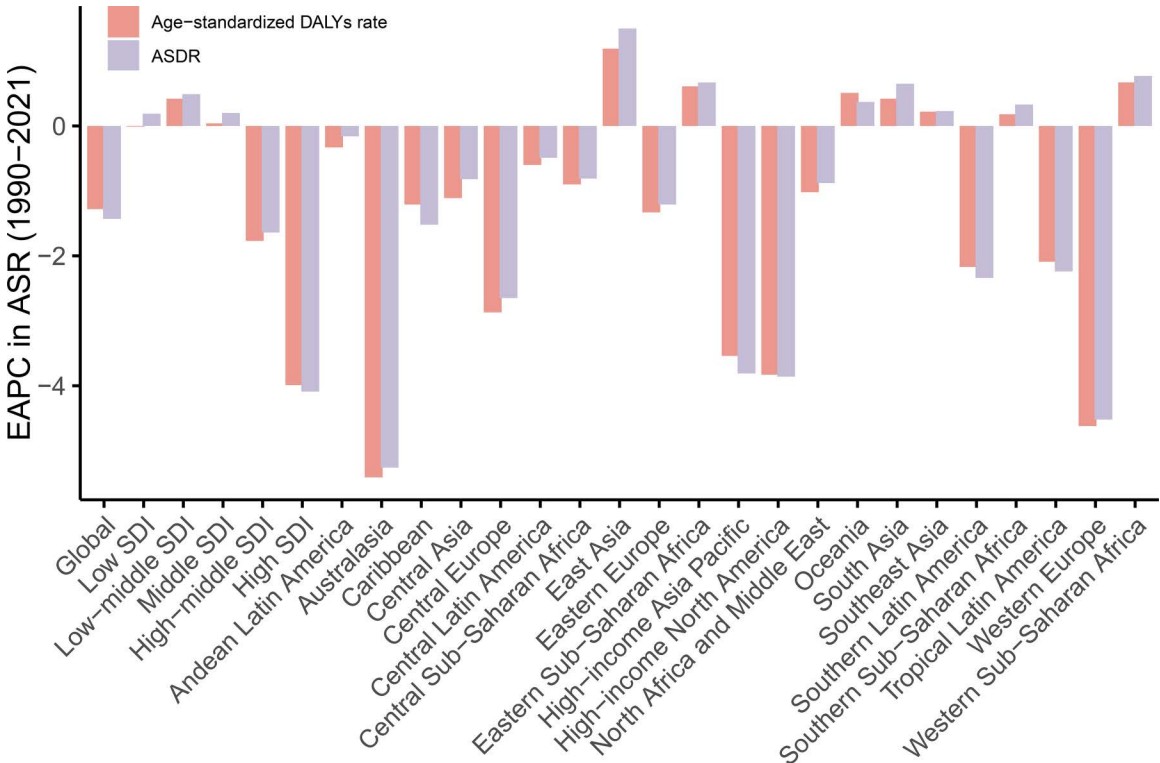

**Fig 3. Estimated annual percentage changes in age-standardized rates of DALYs and ASDR of IHD attributable to HSBP in asthma across 21 regions from 1990 to 2021.**

These findings underscore the urgency of developing and implementing targeted, age- and region-specific strategies to mitigate the growing cardiovascular impact of HSBP worldwide.

## 1. Age- and gender-specific mechanisms and implications

Our findings indicate that the burden of IHD attributable to HSBP is disproportionately concentrated among older adults, with both DALYs and mortality rates increasing progressively with age. This trend reflects the cumulative physiological impact of sustained elevated blood pressure, which accelerates vascular and myocardial remodeling—manifested as endothelial dysfunction, arterial stiffness, and left ventricular hypertrophy—all of which contribute to impaired myocardial perfusion and increased ischemic risk in the elderly population[10,28].

Sex-based differences were also evident. Across nearly all age categories, males experienced higher DALYs and mortality rates than females. This disparity likely reflects both biological and behavioral factors. Estrogen is known to confer vascular protection through mechanisms such as improved nitric oxide availability, reduced oxidative stress, and favorable lipid profiles[29]. These effects may explain the delayed onset of IHD in premenopausal women. However, the loss of estrogen post-menopause accelerates vascular aging and increases cardiovascular risk, which may account for the continued rise in DALYs among women aged >95 years, even as the burden among men begins to plateau[30,31]. Beyond hormonal influences, disparities in clinical presentation, risk perception, and healthcare access may further contribute to sex differences. Women are more likely to present with atypical IHD symptoms, potentially delaying diagnosis and reducing treatment rates. In contrast, men have historically exhibited higher rates of behavioral risk factors—such as tobacco use and poor dietary habits—exacerbating their cardiovascular risk profiles[32,33].

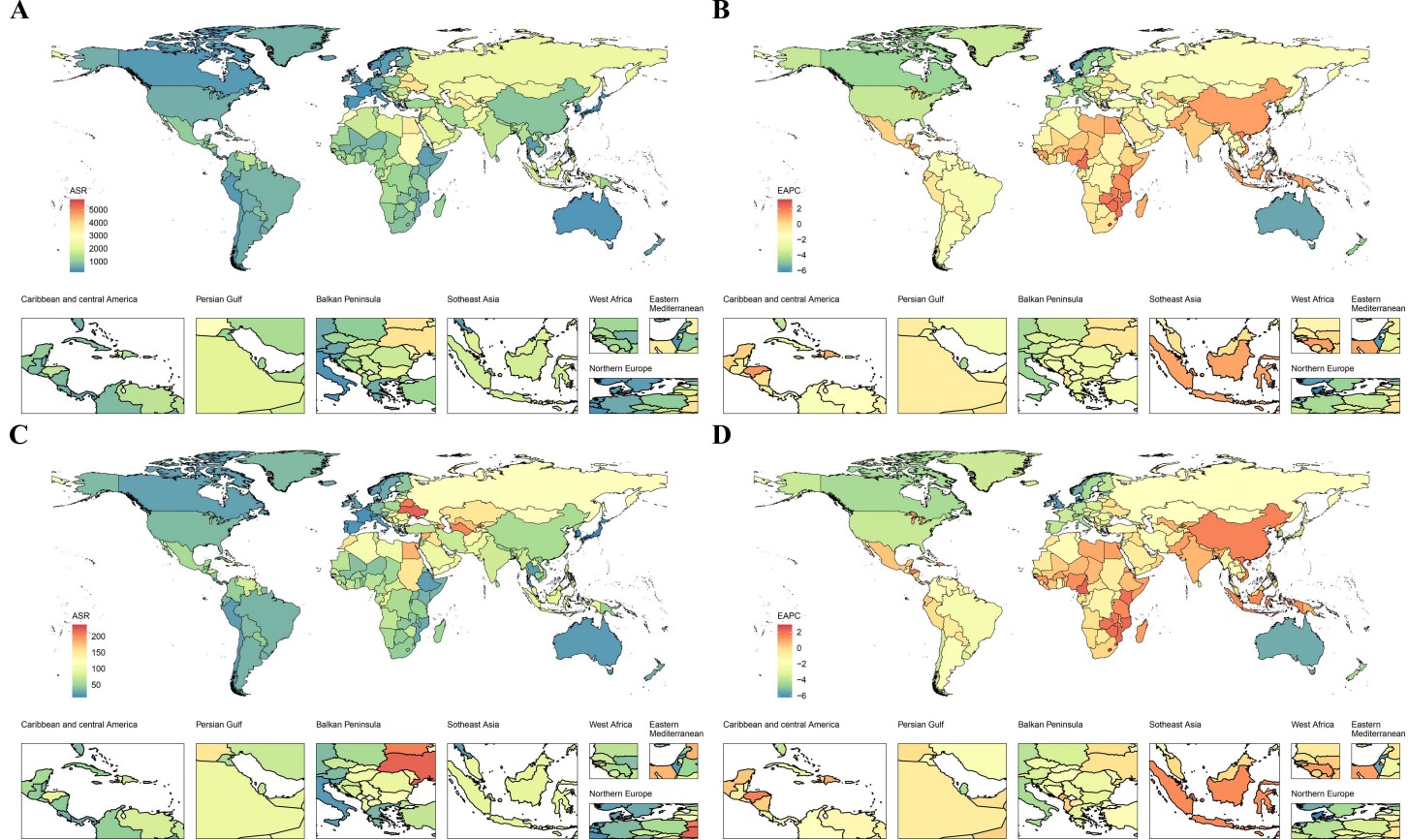

**Fig 4. National age-standardized rates of DALYs and ASDR of IHD attributable to HSBP and their estimated annual percentage changes in 2021.** (A) National age-standardized DALYs rates and their estimated annual percentage changes (EAPC) for IHD attributable to HSBP. (B) National age-standardized death rates and their estimated annual percentage changes (EAPC) for IHD attributable to HSBP. DALYs: Disability-adjusted life years; ASR: Age-standardized rate; EAPC: Estimated annual percentage change. This map was created with permission from Xin-Liang Xu, using map resources at 'https://www.resdc.cn/data.aspx?DATAID=205', original copyright 2025.

Importantly, age-stratified EAPC trends reveal a rising burden among younger adults (25–44 years) in regions such as East Asia, South Asia, and Sub-Saharan Africa, where positive EAPCs contrast with global declines. This shift likely reflects the rapid urbanization, dietary transitions, sedentary lifestyles, and limited hypertension awareness or treatment access among younger populations. These patterns highlight the need for expanded public health efforts beyond elderly populations, emphasizing age-appropriate primary prevention for younger adults in transitioning regions[34,35].

These findings underscore the necessity of tailoring IHD prevention and hypertension control strategies by age and sex. Interventions such as earlier screening in men, aggressive risk factor modification in postmenopausal women, and community-based education campaigns targeting young adults could help reduce the growing disparities in IHD burden attributable to HSBP.

## 2. Regional variations in the burden of HSBP-attributable IHD

Our findings highlight substantial disparities in the burden of IHD attributable to HSBP across different SDI regions, shaped by variations in healthcare infrastructure, policy implementation, hypertension awareness, and socioeconomic conditions. High-SDI countries have made significant progress in hypertension prevention and management due to

**Fig 5. Age-specific estimated annual percentage changes in DALYs and death rates for 21 regions from 1990 to 2021.** (A) Age-specific estimated annual percentage changes in DALYs rates. (B) Age-specific estimated annual percentage changes in death rates. Red represents positive values and blue represents negative values, and the color shades correspond to the magnitude of the values, reflecting the trend and magnitude of changes in each region at the corresponding time.

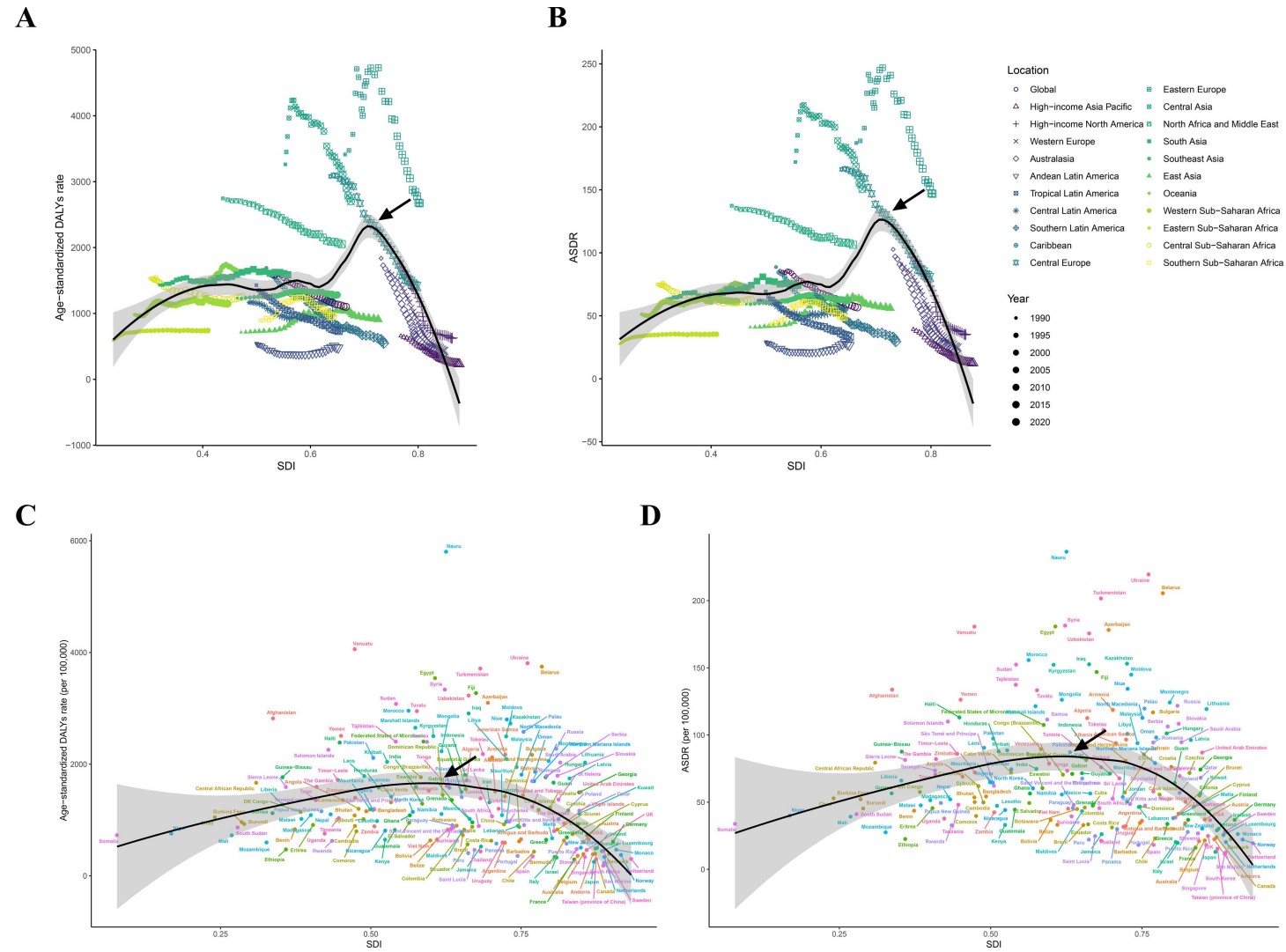

**Fig 6. SDI-based age-standardized rates of DALYs and ASDR for IHD attributable to HSBP.** (A) SDI-based age-standardized rates of DALYs in 21 regions from 1990 to 2021. (B) SDI-based ASDR in 21 regions from 1990 to 2021. (C) SDI-based age-standardized DALYs across countries in 2021. (D) SDI-based ASDR across countries in 2021. Arrows point to major inflection points (SDI≈0.7) where IHD burden begins to decline with increasing SDI. SDI: Socio-demographic Index; DALYs: Disability-adjusted life years; ASR: Age-standardized rate.

well-established healthcare systems, widespread screening programs, and adherence to updated guidelines. Large-scale initiatives, including community-based screening, sodium reduction policies, and government-subsidized antihypertensive medications, have contributed to declining IHD mortality[36]. However, recent studies indicate that hypertension control rates in these regions have plateaued or declined[37], highlighting the need for policy refinements focused on long-term adherence, digital health integration, and reducing residual cardiovascular risk.

In contrast, low-SDI regions continue to exhibit rising ASRs. This increasing trend can be attributed to several health-care and socioeconomic factors. In many low- and middle-SDI countries, limited healthcare infrastructure and weak primary care systems result in low hypertension detection and treatment rates[38]. Additionally, financial barriers to antihypertensive medications, including high out-of-pocket costs and inadequate insurance coverage, reduce treatment

adherence[39]. Even when treatment is available, low health literacy and limited public awareness contribute to late diagnosis and inadequate blood pressure control[38,40]. Moreover, dietary shifts driven by urbanization, characterized by high sodium intake and increased consumption of processed foods, have exacerbated hypertension prevalence in these regions[41,42].

To effectively reduce the burden of HSBP-related IHD, region-specific strategies should be prioritized. High-SDI countries should focus on sustaining and improving hypertension control efforts by addressing stagnation in treatment adherence and integrating digital health technologies for remote blood pressure monitoring[43]. Additionally, policies aimed at reducing residual cardiovascular risk—such as comprehensive risk-factor management programs targeting hyperlipidemia, obesity, and smoking cessation—should be reinforced[44,45]. In contrast, middle- and low-SDI regions must prioritize improving access to cost-effective antihypertensive medications through government subsidies and essential medicine programs. Expanding community-based screening programs and integrating hypertension management into primary healthcare services can significantly enhance detection and treatment rates. Task-shifting models, in which trained non-physician healthcare workers manage hypertension, have shown effectiveness in resource-limited settings[46]–[48]. Furthermore, mobile health (mHealth) interventions, such as SMS-based medication reminders and remote monitoring applications, have improved hypertension management in underserved areas[49,50].

To address these disparities, high-SDI regions should focus on reversing the stagnation in hypertension control rates through improved treatment adherence and digital health tools. Middle- and low-SDI regions should prioritize expanding access to cost-effective antihypertensive medications and integrating hypertension management into primary care[46]. Globally, population-wide strategies such as reducing dietary sodium intake, promoting physical activity, and enforcing tobacco control policies remain critical in mitigating the cardiovascular risks associated with HSBP[39,45,51,52].

## 3. Limitations and clinical significance

This study has several limitations that warrant consideration. First, the GBD framework defines HSBP using a theoretical minimum risk exposure level (TMREL) of 105–115 mmHg, rather than the fixed diagnostic thresholds employed in conventional clinical guidelines. While this continuous modeling approach facilitates more granular risk estimation across the full spectrum of systolic blood pressure levels, it may lead to an overestimation of attributable burden compared to definitions based on clinical hypertension thresholds. As such, our findings should be interpreted as reflecting cardiovascular risk associated with elevated SBP, rather than as direct estimates of clinical hypertension burden. Second, the GBD framework integrates data from heterogeneous sources—including national surveys, hospital records, and epidemiological studies—with varying degrees of completeness and accuracy. Although advanced statistical modeling techniques (e.g., DisMod-MR 2.1, Bayesian inference, and spatiotemporal Gaussian regression) are employed to harmonize data and impute missing values, regional inconsistencies in data collection practices, blood pressure measurement protocols, and diagnostic criteria may introduce systematic bias. Furthermore, underreporting of ischemic heart disease in resource-constrained settings and variability in hypertension assessment methodologies across countries could compromise the accuracy of burden estimates. Third, the cross-sectional design of this study limits causal inference. It is not possible to definitively establish whether elevated SBP preceded the development of IHD or whether observed associations were confounded by preexisting comorbid conditions. Moreover, the ecological nature of GBD estimates—derived from population-level rather than individual-level data—further limits the ability to draw causal inferences across countries or subgroups. Although the comparative risk assessment framework used in the GBD study adjusts for overlapping risk exposures, important confounders such as smoking, obesity, diabetes, and dietary patterns were not explicitly accounted for, leaving room for residual confounding. Finally, the use of national-level aggregates—such as SDI and blood pressure data—may obscure substantial within-country disparities, particularly in large or socioeconomically diverse countries. For example, individuals in low-SDI countries may, in some contexts, have access to high-quality healthcare, while marginalized groups in high-SDI countries may face significant barriers to hypertension diagnosis and treatment. Future research

should incorporate individual-level and longitudinal data to improve risk estimation and better elucidate the complex interactions among HSBP, socioeconomic status, and IHD outcomes. Additionally, while no formal sensitivity analyses were performed, the 95% uncertainty intervals generated by the GBD modeling framework offer an indication of variability around the estimates.

Despite these limitations, our analysis provides meaningful insights into the demographic and geographic variability of IHD burden attributable to hypertension. By identifying high-risk populations and regions, the findings offer a basis for targeted interventions and policy development. The observed parabolic relationship between SDI and disease burden suggests that uniform strategies may be suboptimal. In lower-SDI settings, efforts should focus on early detection, treatment accessibility, and primary care integration. In higher-SDI contexts, optimizing adherence, promoting healthy lifestyles, and addressing care gaps remain key. Aligning hypertension control strategies with health system capacity and socioeconomic conditions may help reduce global disparities in cardiovascular outcomes.

## Conclusion

This study highlights the growing global burden of HSBP-related IHD, with significant disparities across age groups, genders, and regions. While age-standardized rates have declined, the absolute burden remains high, particularly in low- and middle-SDI regions, where limited healthcare access and low treatment adherence drive rising trends. To address this, high-SDI regions should focus on improving long-term hypertension control through digital health tools and adherence strategies, while low- and middle-SDI regions must prioritize expanding access to affordable antihypertensive medications and integrating hypertension management into primary care. Future research should assess the effectiveness of targeted interventions and explore policy-driven strategies to enhance hypertension prevention and treatment, ultimately reducing IHD burden worldwide.

## Supporting information

**S1 Table.  The rates of DALYs and deaths of ischemic heart disease attributable to high systolic blood pressure in 2021.**
(XLSX)

**S2 Table.  The 2021 Age-standardised rate of DALYs and deaths for ischemic heart disease attributable to high systolic blood pressure in 204 countries and territories.**
(XLSX)

**S1 File.  R code for EAPC calculation and correlation analysis of ASRs with SDI.**
(TXT)

## Acknowledgments

We truly appreciate the efforts of the Global Burden of Disease study 2021 collaborators in delivering the most complete study of various diseases on a worldwide scale.

## Author contributions

**Conceptualization:** Zhenhua Zhuang, Peng Guo.

**Data curation:** Zhenhua Zhuang, Peng Guo.

**Formal analysis:** Zhenhua Zhuang, Haifeng Li.

**Investigation:** Zhenhua Zhuang, Qiuju Wang, Shenghong Lan, Yanru Su.

**Methodology:** Zhenhua Zhuang, Qiuju Wang, Yanru Su, Yue Lin.

**Project administration:** Yue Lin, Peng Guo.

**Software:** Zhenhua Zhuang, Haifeng Li.

**Supervision:** Yue Lin, Peng Guo.

**Visualization:** Shenghong Lan, Yanru Su, Yue Lin.

**Writing – original draft:** Zhenhua Zhuang, Qiuju Wang.

**Writing – review & editing:** Yue Lin, Peng Guo.

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
