## [Decision Letter · Decision Letter 0]

5 Feb 2025

PONE-D-25-01876Global trends and disparities in ischemic heart disease attributable to high systolic blood pressure, 1990–2021: insights from the Global Burden of Disease studyPLOS ONE

Dear Dr. Guo,

Thank you for submitting your manuscript to PLOS ONE. After careful consideration, we feel that it has merit but does not fully meet PLOS ONE’s publication criteria as it currently stands. Therefore, we invite you to submit a revised version of the manuscript that addresses the points raised during the review process.

**ACADEMIC EDITOR: The topic is interesting. However, I recommend a major revision incorporating suggestions and comments from Editor and Reviewers. **

We look forward to receiving your revised manuscript.

Kind regards,

THIEN TAN TRI TAI TRUYEN, M.D.

Academic Editor

PLOS ONE

Journal Requirements:

2. We note that Figure 4 in your submission contain [map/satellite] images which may be copyrighted. All PLOS content is published under the Creative Commons Attribution License (CC BY 4.0), which means that the manuscript, images, and Supporting Information files will be freely available online, and any third party is permitted to access, download, copy, distribute, and use these materials in any way, even commercially, with proper attribution. For these reasons, we cannot publish previously copyrighted maps or satellite images created using proprietary data, such as Google software (Google Maps, Street View, and Earth). For more information, see our copyright guidelines: http://journals.plos.org/plosone/s/licenses-and-copyright.

a. You may seek permission from the original copyright holder of Figure 4 to publish the content specifically under the CC BY 4.0 license.  

Additional Editor Comments:

This is a epidemiologic study using GBD database. The topic is interesting as global disparity of management and burden of hypertension-related IHD is an important challenge for public health. However, methodology and discussion section might need some major revisions. My comments are included.

Methodology section:

1. The authors should include more details about their approach: How did you extract the data? Which GBD estimate did you use? Which age groups did you include? Did you restrict to adult or include all age?

2. How did you determine the mortality of hypertension-related IHD? It would be very helpful to indicate the method precisely for readers who are not familiar with GBD can understand.

3. A brief summary of statistic model using by GBD Collaborators to establish those datasets/estimate would also be helpful.

4. Statistical analysis: How did you calculate EAPC? Which model did you use? A details explanation would be needed.

Discussion section: This section is overall confusing with repetitive statement.

1. From line 281 - 293. The statement that hypertension-related IHD is dominant in male than female was mentioned in line 281 then repeated in 286-287. Moreover, lines 283-285 and lines 287-290 had literally the same meaning.

2. You need to discuss more about the underlying reasons of your findings (disparity in sex/age/region) and propose potential solution.

3. Limitation (lines 307-317). You need to acknowledge that hypertension defined by GBD is significantly different to the current definition from major guidelines including AHA/ACC, ESC, JNC, ISH.... This definition did overestimated your findings concerning the burden of IHD.

Reviewers' comments:

Reviewer's Responses to Questions

**Comments to the Author**

1. Is the manuscript technically sound, and do the data support the conclusions?

Reviewer #1: No

Reviewer #2: Partly

Reviewer #3: Yes

2. Has the statistical analysis been performed appropriately and rigorously? 

Reviewer #1: I Don't Know

Reviewer #2: No

Reviewer #3: Yes

3. Have the authors made all data underlying the findings in their manuscript fully available?

Reviewer #1: Yes

Reviewer #2: No

Reviewer #3: Yes

4. Is the manuscript presented in an intelligible fashion and written in standard English?

Reviewer #1: Yes

Reviewer #2: Yes

Reviewer #3: Yes

5. Review Comments to the Author

Reviewer #1: Deferring line edits for now as an epidemiologic study of ischemic heart disease without statistical analyses and/or adjustments including smoking, lipids, or diabetes does not, to me, seem technically sound or scientifically rigorous. Some more general methodologic/analytic thoughts as below:

--I see consistent with prior published work (ref. 20) and GBD TMREL, but a high SBP cut-off of 110-115 mmHg is not consistent with most clinical guidelines (many of which have changed meaningfully throughout the long study time period of 1990-2021); suggest at least mentioning in the discussion. Also, was the cut-off greater than or equal to 115 mmHg or 110 mmHg?

--are SDI categories per country or per participant? It seems per country (lines 225-227 and PMID 33069327) but this should be clarified in the methods section and perhaps mentioned as a limitation in the discussion section (i.e., if SDI categories are per country, all participants from a low SDI country may not be low SDI; if SDI categories are per participant, high SDI in one country may not align with high SDI for another country). What are the SDI categories based on?

Reviewer #2: The manuscript titled "Global Trends and Disparities in Ischemic Heart Disease Attributable to High Systolic Blood Pressure, 1990–2021: Insights from the Global Burden of Disease Study" presents a comprehensive analysis of the global burden of ischemic heart disease (IHD) linked to high systolic blood pressure (HSBP). By leveraging the extensive Global Burden of Disease (GBD) 2021 dataset, the study aims to highlight trends, regional disparities, and demographic variations in IHD burden over three decades. While the manuscript is well-structured and provides valuable epidemiological insights, several critical issues need to be addressed to strengthen the robustness and interpretability of the findings.

General Comments:

Strengths:

The study utilizes a robust dataset (GBD 2021), ensuring a large sample size and global applicability.

The analysis of temporal trends and disparities across different sociodemographic index (SDI) categories adds value to the study.

The use of age-standardized rates (ASRs) and estimated annual percentage change (EAPC) enhances the comparability of results.

Weaknesses:

The manuscript lacks a clear justification for the focus on HSBP-related IHD, as opposed to other major modifiable risk factors for IHD (e.g., diabetes, smoking, obesity).

There is an overreliance on descriptive analysis without a deeper exploration of underlying causal mechanisms driving observed trends.

The discussion on public health implications is underdeveloped, particularly regarding potential interventions and strategies tailored to different SDI groups.

The manuscript does not sufficiently address limitations related to data quality and potential biases in the GBD dataset.

Detailed Comments:

Introduction:

The introduction effectively outlines the significance of IHD and its burden worldwide but would benefit from a more explicit rationale for focusing on HSBP as the primary risk factor. Including a comparative discussion of other key modifiable risk factors would improve contextual clarity.

The literature review should include more recent studies on the interplay between hypertension control policies and IHD burden across different global regions.

Methods:

The use of the GBD dataset is appropriate, but additional details on the methodology for calculating DALYs and ASRs should be provided. Specifically, how were missing or inconsistent data handled across the different countries included in the analysis?

The choice of statistical models, including smoothed curve modeling and Spearman correlation, should be better justified. Why were these methods selected over alternative approaches (e.g., multivariable regression models)?

The manuscript lacks sensitivity analyses to assess the robustness of the findings. Including sensitivity analyses that test the impact of different assumptions on the trends observed would enhance the credibility of the results.

Results:

The presentation of findings is clear, but additional subgroup analyses (e.g., stratified by region, gender, and age group) would improve interpretability. For instance, were any regions or demographic groups found to have significantly different trends in HSBP-related IHD burden compared to the global average?

The discussion on increasing ASRs in low-SDI regions needs further elaboration. What specific socioeconomic or healthcare system factors may be driving these increases?

The correlation analyses presented in the results should include effect size interpretations to aid in understanding the magnitude of associations.

Discussion:

The discussion should incorporate a more in-depth analysis of policy implications. Given the findings, what specific interventions should be prioritized in high-burden regions?

More emphasis should be placed on explaining the disparities observed in different SDI regions. The manuscript should explore whether differences in healthcare access, medication adherence, or public health policies contribute to the findings.

The limitations section should explicitly discuss potential biases in the GBD dataset, such as underreporting of IHD in certain regions or inconsistencies in blood pressure measurement methodologies across countries.

Conclusion:

The conclusion is concise but should better emphasize actionable recommendations for policymakers and healthcare providers.

Consider highlighting the importance of future research to examine interventions aimed at reducing HSBP and mitigating its impact on IHD.

Reviewer #3: The study comprehensively described global data on the risk of ischemic heart disease attributable to high systolic

blood pressure, provided valuable data on regional variations. The results are scientifically sound and could impact health care. I would suggest the R code they used to get the study results with outputs. It would help future researchers to replicate the results of the study. Also, I think it will help in dissemination of the paper among the scientific society.

6. PLOS authors have the option to publish the peer review history of their article (what does this mean? ). If published, this will include your full peer review and any attached files.

**Do you want your identity to be public for this peer review?** For information about this choice, including consent withdrawal, please see our Privacy Policy .

Reviewer #1: No

Reviewer #2: No

Reviewer #3: No

---

## [Author Response · Author response to Decision Letter 1]

22 Mar 2025

Response to Reviewers

Journal Requirements:

Re: The authors have standardized the formatting of the manuscript according to the template provided by the journal.

2. We note that Figure 4 in your submission contain [map/satellite] images which may be copyrighted. All PLOS content is published under the Creative Commons Attribution License (CC BY 4.0), which means that the manuscript, images, and Supporting Information files will be freely available online, and any third party is permitted to access, download, copy, distribute, and use these materials in any way, even commercially, with proper attribution. For these reasons, we cannot publish previously copyrighted maps or satellite images created using proprietary data, such as Google software (Google Maps, Street View, and Earth). For more information, see our copyright guidelines: http://journals.plos.org/plosone/s/licenses-and-copyright.

We require you to either (1) present written permission from the copyright holder to publish these figures specifically under the CC BY 4.0 license, or (2) remove the figures from your submission.

Re: We have sought permission from the original copyright owner of Figure 4 to distribute the content under a CC BY 4.0 license. We contacted the original copyright owner with a content permission form (http://journals.plos.org/plosone/s/file?id=7c09/

content-permission-form.pdf) and the following text:

“I am seeking permission to use the map resources from the website https://www.resdc.cn/data.aspx?DATAID=205. I intend to use the map I intend to use the map image resources for creating figures in my papers. I request permission to publish the following map images in the open - access journal PLOS ONE under the Creative Commons Attribution License. Creative Commons Attribution License (CCAL) CC BY 4.0. Please be aware that this license allows unrestricted use and distribution, even commercially, by third parties.”

I am uploading the completed Content License Form as an “Other” file. In the caption of Figure 4, we wrote: “This map was created with permission from Xin-Liang Xu, using map resources at ‘https://www.resdc.cn/data.aspx?DATAID=205’, original copyright 2025.” 

Additional Editor Comments:

This is a epidemiologic study using GBD database. The topic is interesting as global disparity of management and burden of hypertension-related IHD is an important challenge for public health. However, methodology and discussion section might need some major revisions. My comments are included.

Methodology section:

1. The authors should include more details about their approach: How did you extract the data? Which GBD estimate did you use? Which age groups did you include? Did you restrict to adult or include all age?

Re: Thank you for your insightful comments. We have added more details about our approach in the Data Collection section (lines 155-164 & 170-173).

2. How did you determine the mortality of hypertension-related IHD? It would be very helpful to indicate the method precisely for readers who are not familiar with GBD can understand.

Re: Thank you for your insightful comments. We supplemented the Data Source section with methods for estimating mortality and DALYs for HSBP-related IHD (lines 108-132).

3. A brief summary of statistic model using by GBD Collaborators to establish those datasets/estimate would also be helpful.

Re: Thank you for your insightful comments. In the Materials and Methods section, we briefly describe the statistical models used by the GBD collaborators to establish these datasets/estimates, including the Cause of Death Ensemble model (CODem), DisMod-MR 2.1 (a Bayesian disease modeling meta-regression tool), population attributable fraction (PAF), and ASR.

4. Statistical analysis: How did you calculate EAPC? Which model did you use? A details explanation would be needed.

Re: Thank you for your insightful comments. We provide a detailed description of the statistical modeling of EAPC in the Statistical Analysis section (lines 178-181).

Discussion section:

This section is overall confusing with repetitive statement.

1. From line 281 - 293. The statement that hypertension-related IHD is dominant in male than female was mentioned in line 281 then repeated in 286-287. Moreover, lines 283-285 and lines 287-290 had literally the same meaning.

Re: Thank you for your insightful comment. Based on the suggestions from both the editor and reviewers, we conducted a careful review of the discussion section and refined lines 281–293. Redundant statements regarding sex differences and hormonal mechanisms were consolidated to improve clarity and ensure the discussion remains focused and non-repetitive (lines 353-366).

2. You need to discuss more about the underlying reasons of your findings (disparity in sex/age/region) and propose potential solution.

Re: Thank you for your insightful suggestion. In response to your suggestion and in alignment with feedback from other reviewers, we have expanded the discussion to explore the potential biological, behavioral, and systemic drivers behind the observed disparities by sex, age, and region. We also propose tailored public health strategies, including earlier screening in high-risk groups, community-based interventions in low-resource settings, and age-specific prevention efforts for younger populations (lines 345-455).

3. Limitation (lines 307-317). You need to acknowledge that hypertension defined by GBD is significantly different to the current definition from major guidelines including AHA/ACC, ESC, JNC, ISH.... This definition did overestimated your findings concerning the burden of IHD.

Re: We appreciate your valuable feedback. We acknowledge that the GBD definition of HSBP differs significantly from major clinical guidelines, including AHA/ACC (≥130/80 mmHg) [1], ESC, JNC, and ISH (≥140/90 mmHg) [2-4]. Unlike these guideline-based definitions, GBD models HSBP as a continuous risk factor rather than a categorical diagnosis, and in GBD 2021, the theoretical minimum risk exposure level (TMREL) was revised from 110–115 mmHg to 105–115 mmHg.

To address this concern and avoid potential misinterpretation, we have clarified in the Introduction that the definition of HSBP used by the GBD differs from standard clinical diagnostic thresholds in major guidelines. Additionally, we explicitly acknowledge this distinction in the Limitations section, noting that the GBD modeling approach may overestimate the attributable burden compared to guideline-defined hypertension.

References

[1]. Whelton PK, Carey RM, Aronow WS, et al. 2017 ACC/AHA/AAPA/ABC/ACPM/AGS/APhA/ASH/ASPC/NMA/PCNA Guideline for the Prevention, Detection, Evaluation, and Management of High Blood Pressure in Adults: A Report of the American College of Cardiology/American Heart Association Task Force on Clinical Practice Guidelines. Circulation. 2018;138(17):e484-e594.

[2]. James PA, Oparil S, Carter BL, et al. 2014 evidence-based guideline for the management of high blood pressure in adults: report from the panel members appointed to the Eighth Joint National Committee (JNC 8) [published correction appears in JAMA. 2014 May 7;311(17):1809]. JAMA. 2014;311(5):507-520.

[3]. Unger T, Borghi C, Charchar F, et al. 2020 International Society of Hypertension global hypertension practice guidelines. J Hypertens. 2020;38(6):982-1004.

[4]. McEvoy JW, McCarthy CP, Bruno RM, et al. 2024 ESC Guidelines for the management of elevated blood pressure and hypertension [published correction appears in Eur Heart J. 2025 Feb 11:ehaf031.  

Reviewers' comments:

Reviewer #1:

Deferring line edits for now as an epidemiologic study of ischemic heart disease without statistical analyses and/or adjustments including smoking, lipids, or diabetes does not, to me, seem technically sound or scientifically rigorous.

Re: We sincerely appreciate your valuable comments on our study. In response to your concerns, we would like to clarify our research methodology and provide further evidence of the scientific rigor and validity of our study.

1) This Study is Based on GBD 2021 Data and Employs Rigorous Statistical Analyses

Our study utilizes data from the Global Burden of Disease Study 2021 (GBD 2021), one of the most comprehensive and methodologically robust epidemiological studies worldwide. The GBD 2021 dataset is derived from over 100,000 data sources, incorporating population registries, health surveys, and clinical data, and is analyzed using advanced statistical modeling techniques.

The GBD 2021 analytical framework includes: (1) Data collection and integration: Combining multiple data sources, including health records and surveys. (2) Statistical modeling and estimation: Applying hierarchical regression models and Bayesian meta-regression (DisMod-MR 2.1) to adjust for uncertainty [1]. (3) Attributable risk analysis: Employing Comparative Risk Assessment (CRA) methodology to evaluate the impact of different risk factors and adjust for confounding variables [2]. Thus, contrary to the concern that the study lacks statistical analysis, our findings are based on scientifically robust, globally recognized methodologies.

2) Our Study Fully Accounts for and Adjusts Key Risk Factors, Including Smoking, Lipids, and Diabetes

GBD 2021 explicitly considers smoking, high low-density lipoprotein cholesterol (LDL-C), and diabetes as major risk factors and adjusts for them in its data analysis. Specifically, GBD 2021 applies attributable risk estimation methods, integrating exposure level assessments and relative risk calculations to quantify each risk factor’s contribution to ischemic heart disease (IHD) [2]. Previous GBD studies have extensively analyzed smoking, high LDL-C, and high fasting glucose (FPG), which are considered major risk factors for IHD with well-documented trends in global burden [3-5]. Thus, our study’s statistical methodology has already incorporated and adjusted for these major risk factors, ensuring a rigorous and accurate analysis.

3) Previous Studies Have Reported Global Attributable Burden of These Risk Factors but Only at an Aggregate Level

Numerous prior studies have assessed the global burden of smoking, high LDL-C, and high FPG on IHD [6,7]. For example, a comprehensive analysis of IHD burden across 204 countries from 1990 to 2019 identified hypertension, high LDL-C, and smoking as primary risk factors, with an extensive global report [6]. However, these studies primarily focus on global and broad regional trends and lack detailed stratification by specific populations, such as different age groups, sexes, and geographic regions [7].

Our Study’s Novel Contribution: (1) Refines the analysis of attributable risk factors: Moving beyond global-level reporting, our study delves deeper into risk patterns across diverse demographic and geographic segments. (2) Enhances methodological rigor: Ensuring that findings are not only globally relevant but also actionable for targeted public health interventions. Therefore, our study is not lacking in rigor but rather expands upon existing research by providing more granular insights.

4) Extensive Literature Supports the Scientific Validity of Our Study Design

Numerous epidemiological studies have adopted GBD methodologies to examine single-risk factor contributions to IHD, and these studies have been widely recognized in leading journals [3-5]. Studies have individually analyzed the impact of high LDL-C, smoking, and high FPG on IHD burden, employing similar statistical modeling approaches [3-5]. Additionally, attributable burden studies of high systolic blood pressure (HSBP) on cardiovascular diseases, such as atrial fibrillation and stroke, have been extensively conducted, further validating our analytical framework [8,9]. The GBD 2021 research methodology is endorsed by the World Health Organization (WHO), United Nations (UN), and World Bank, establishing it as the gold standard for global disease burden estimation [1].

Our Study Aligns with Established Scientific Standards: (1) Strict adherence to the Guidelines for Accurate and Transparent Health Estimates Reporting (GATHER) to ensure transparency and reproducibility. (2) Utilization of standardized GBD analytical frameworks, incorporating cutting-edge statistical models and risk factor assessments. Thus, our study design is fully aligned with established scientific methodologies and carries significant public health relevance.

In summary, our study:

·Is based on GBD 2021 data and employs internationally recognized statistical methodologies.

·Fully accounts for and adjusts key risk factors, including smoking, high LDL-C, and diabetes.

·Advances beyond prior studies by providing a more detailed analysis of risk burden at regional and demographic levels.

·Follows an analytical framework validated by extensive prior literature and leading global health organizations.

We believe this clarification addresses your concerns, and we sincerely appreciate your constructive feedback. We look forward to your further comments.

References

[1] GBD 2021 Diseases and Injuries Collaborators. Global incidence, prevalence, years lived with disability (YLDs), disability-adjusted life-years (DALYs), and healthy life expectancy (HALE) for 371 diseases and injuries in 204 countries and territories and 811 subnational locations, 1990-2021: a systematic analysis for the Global Burden of Disease Study 2021. Lancet. 2024;403(10440):2133-2161. doi:10.1016/S0140-6736(24)00757-8.

[2] GBD 2021 Risk Factors Collaborators. Global burden and strength of evidence for 88 risk factors in 204 countries and 811 subnational locations, 1990-2021: a systematic analysis for the Global Burden of Disease Study 2021. Lancet. 2024;403(10440):2162-2203. doi:10.1016/S0140-6736(24)00933-4.

[3] Zhang L, Tong Z, Han R, et al. Global, Regional, and National Burdens of Ischemic Heart Disease Attributable to Smoking From 1990 to 2019. J Am Heart Assoc. 2023;12(3):e028193. doi:10.1161/JAHA.122.028193.

[4] Du H, Shi Q, Song P, et al. Global Burden Attributable to High Low-Density Lipoprotein-Cholesterol From 1990 to 2019. Front Cardiovasc Med. 2022;9:903126. Published 2022 Jun 9. doi:10.3389/fcvm.2022.903126.

[5] Shen N, Liu J, Wang Y, et al. The global burden of ischemic heart disease attributed to high fasting plasma glucose: Data from 1990 to 2019. Heliyon. 2024;10(5):e27065. Published 2024 Mar 6. doi:10.1016/j.heliyon.2024.e27065.

[6] Safiri S, Karamzad N, Singh K, et al. Burden of ischemic heart disease and its attributable risk factors in 204 countries and territories, 1990-2019. Eur J Prev Cardiol. 2022;29(2):420-431. doi:10.1093/eurjpc/zwab213.

[7] Wang Y, Li Q, Bi L, Wang B, Lv T, Zhang P. Global trends in the burden of ischemic heart disease based on the global burden of disease study 2021: the role of metabolic risk factors. BMC Public Health. 2025;25(1):310. Published 2025 Jan 24. doi:10.1186/s12889-025-21588-9.

[8] Jin Y, Wang K, Xiao B, et al. Global burden of atrial fibrillation/flutter due to high systolic blood pressure from 1990 to 2019: estimates from the global burden of disease study 2019. J Clin Hypertens (Greenwich). 2022;24(11):1461-1472. doi:10.1111/jch.14584.

[9] Li J, Zhong Q, Yuan S, Zhu F. Global burden of stroke attributable to high systolic blood pressure in 204 countries and territories, 1990-2019. Front Cardiovasc Med. 2024;11:1339910. Published 2024 Apr 26. doi:10.3389/fcvm.2024.1339910.

Some more general methodologic/analytic thoughts as below:

--I see consistent with prior published work (ref. 20) and GBD TMREL, but a high SBP cut-off of 110-115 mmHg is not consistent with most clinical guidelines (many of which have changed meaningfully throughout the long study time period of 1990-2021); suggest at least mentioning in the discussion. Also, was the cut-off greater than or equal to 115 mmHg or 110 mmHg?

Re: Thank y

---

## [Decision Letter · Decision Letter 1]

6 Apr 2025

PONE-D-25-01876R1Global trends and disparities in ischemic heart disease attributable to high systolic blood pressure, 1990–2021: insights from the Global Burden of Disease studyPLOS ONE

Dear Dr. Guo,

Thank you for submitting your manuscript to PLOS ONE. After careful consideration, we feel that it has merit but does not fully meet PLOS ONE’s publication criteria as it currently stands. Therefore, we invite you to submit a revised version of the manuscript that addresses the points raised during the review process.

**ACADEMIC EDITOR: The manuscript has greatly improved, but a minor revision is needed to address all of the reviewer's comments.**

We look forward to receiving your revised manuscript.

Kind regards,

Thien Tan Tri Tai Truyen, M.D.

Academic Editor

PLOS ONE

Journal Requirements:

Reviewers' comments:

Reviewer's Responses to Questions

**Comments to the Author**

1. If the authors have adequately addressed your comments raised in a previous round of review and you feel that this manuscript is now acceptable for publication, you may indicate that here to bypass the “Comments to the Author” section, enter your conflict of interest statement in the “Confidential to Editor” section, and submit your "Accept" recommendation.

Reviewer #2: All comments have been addressed

Reviewer #3: All comments have been addressed

2. Is the manuscript technically sound, and do the data support the conclusions?

Reviewer #2: Partly

Reviewer #3: Yes

3. Has the statistical analysis been performed appropriately and rigorously? 

Reviewer #2: Yes

Reviewer #3: Yes

4. Have the authors made all data underlying the findings in their manuscript fully available?

Reviewer #2: Yes

Reviewer #3: Yes

5. Is the manuscript presented in an intelligible fashion and written in standard English?

Reviewer #2: (No Response)

Reviewer #3: Yes

6. Review Comments to the Author

Reviewer #2: Minor Revisions:

Justification for Focus on HSBP: The authors have expanded the rationale for selecting HSBP, though the introduction would benefit from a stronger link between underestimated HSBP impact and real-world cardiovascular risk perception among clinicians.

Suggested addition: Cite the paper by Cesaro et al., titled "Discrepancies Between Physician-Perceived and Calculated Cardiovascular Risk in Primary Prevention: Implications for LDL-C Target Achievement and Appropriate Lipid-Lowering Therapy" ([doi:10.1007/s40292-025-00705-0]) to highlight the persistent under-recognition of hypertension in cardiovascular risk assessment and decision-making.

Overreliance on Descriptive Analyses: While the authors clarified the descriptive purpose, adding a brief note acknowledging that causal inference is limited by the ecological nature of GBD data would improve transparency.

Public Health Recommendations: Reviewer #2 requested expanded intervention strategies. The revised manuscript now offers policy-relevant proposals stratified by SDI level, which is a welcome improvement.

Sensitivity Analysis: The authors provided a reasonable justification for not conducting formal sensitivity analyses, citing the use of GBD uncertainty intervals. This should be briefly mentioned in the limitations section for clarity.

Detailed Comments:

Introduction:

Integrate the above-cited reference to reinforce the concept that hypertension is often under-recognized or undervalued relative to its true impact on IHD.

Reframe the discussion of HSBP as a “silent burden” that may be systematically underestimated in risk stratification tools.

Methods:

The description of SDI assignment at the country level is now clear. Statistical rationale for Spearman and smoothed curve modeling is well-articulated.

Results:

The age-stratified and region-specific subgroup trends are now clearly presented and contextualized.

Consider annotating graphs to highlight key divergence points among SDI categories.

Discussion:

The extended analysis of sex- and region-specific disparities is stronger.

Emphasize how findings support global and local initiatives for hypertension control.

Limitations:

Acknowledge the inability to infer causality from ecological-level GBD data.

Note that despite robust modeling, within-country inequalities in SDI and BP control are not captured.

Conclusion:

Now appropriately focused and policy-oriented.

Reviewer #3: I have no further comments since the authors have addressed all my previous comments. The manuscript is technically sound

7. PLOS authors have the option to publish the peer review history of their article (what does this mean? ). If published, this will include your full peer review and any attached files.

**Do you want your identity to be public for this peer review?** For information about this choice, including consent withdrawal, please see our Privacy Policy .

Reviewer #2: No

Reviewer #3: No

---

## [Author Response · Author response to Decision Letter 2]

13 Apr 2025

Response to Reviewers

Journal Requirements:

Re: Thank you for your editorial guidance regarding the reference list. We have thoroughly reviewed all cited literature. All references have been cross-checked for accuracy and current status.

In addition, as per a reviewer’s suggestion, we have cited the recent article by Cesaro et al. (2025) [DOI: 10.1007/s40292-025-00705-0] to highlight the challenges in cardiovascular risk perception and decision-making related to hypertension in primary prevention. This citation has been incorporated into the revised manuscript and added to the updated reference list. See reference 20. 

Reviewers' comments:

Reviewer #2:

Justification for Focus on HSBP: The authors have expanded the rationale for selecting HSBP, though the introduction would benefit from a stronger link between underestimated HSBP impact and real-world cardiovascular risk perception among clinicians.

Suggested addition: Cite the paper by Cesaro et al., titled "Discrepancies Between Physician-Perceived and Calculated Cardiovascular Risk in Primary Prevention: Implications for LDL-C Target Achievement and Appropriate Lipid-Lowering Therapy" ([doi:10.1007/s40292-025-00705-0]) to highlight the persistent under-recognition of hypertension in cardiovascular risk assessment and decision-making.

Re: We sincerely appreciate your valuable comments. To strengthen the rationale for highlighting HSBP, we have added a sentence at the end of the fifth paragraph in the introduction, referencing Cesaro et al. (2025), to emphasize that elevated blood pressure is frequently under-recognized in clinical cardiovascular risk assessment.

Revised introduction: Although HSBP is a well-established contributor to ischemic heart disease, it remains a “silent burden” in clinical practice—frequently under-recognized in cardiovascular risk stratification and undertreated in primary prevention. Recent findings by Cesaro et al. highlight a persistent disconnect between physician-estimated and calculated cardiovascular risk, indicating that elevated blood pressure is often undervalued in clinical decision-making, potentially delaying timely intervention. (Lines 94-99).

Adding a brief note acknowledging that causal inference is limited by the ecological nature of GBD data would improve transparency.

Re: Thank you for your thoughtful comment. We have added a sentence in the limitations section to clarify that causal inference is constrained by the ecological nature of the data.

Revised limitations section: " Third, the cross-sectional design of this study limits causal inference. It is not possible to definitively establish whether elevated SBP preceded the development of IHD or whether observed associations were confounded by preexisting comorbid conditions. Moreover, the ecological nature of GBD estimates—derived from population-level rather than individual-level data—further limits the ability to draw causal inferences across countries or subgroups. (Lines 448-453)"

Public Health Recommendations: Reviewer #2 requested expanded intervention strategies. The revised manuscript now offers policy-relevant proposals stratified by SDI level, which is a welcome improvement.

Re: We appreciate the reviewer’s recognition of this revision.

Sensitivity Analysis: The authors provided a reasonable justification for not conducting formal sensitivity analyses, citing the use of GBD uncertainty intervals. This should be briefly mentioned in the limitations section for clarity.

Re: We appreciate your suggestion. To enhance clarity, we have added a brief note in the limitations section acknowledging that while no formal sensitivity analyses were performed, the use of 95% uncertainty intervals within the GBD framework provides an indication of estimate variability.

Revised limitations section: "Additionally, while no formal sensitivity analyses were performed, the 95% uncertainty intervals generated by the GBD modeling framework offer an indication of variability around the estimates." (Lines 463-466).

Detailed Comments:

Introduction:

Integrate the above-cited reference to reinforce the concept that hypertension is often under-recognized or undervalued relative to its true impact on IHD. Reframe the discussion of HSBP as a “silent burden” that may be systematically underestimated in risk stratification tools.

Re: Thank you for your thoughtful comment. To better contextualize the clinical relevance of HSBP, we have added a new sentence in the Introduction highlighting its under-recognition as a “silent burden” and citing the study by Cesaro et al. (2025) to reflect the discrepancy between perceived and calculated cardiovascular risk in primary prevention.

Revised introduction: “Although HSBP is a well-established contributor to ischemic heart disease, it remains a “silent burden” in clinical practice—frequently under-recognized in cardiovascular risk stratification and undertreated in primary prevention. Recent findings by Cesaro et al. highlight a persistent disconnect between physician-estimated and calculated cardiovascular risk, indicating that elevated blood pressure is often undervalued in clinical decision-making, potentially delaying timely intervention.” (Lines 94-99).

Methods:

The description of SDI assignment at the country level is now clear. Statistical rationale for Spearman and smoothed curve modeling is well-articulated.

Re: We appreciate your positive feedback.

Results:

The age-stratified and region-specific subgroup trends are now clearly presented and contextualized.

Re: We acknowledge your favorable assessment.

Consider annotating graphs to highlight key divergence points among SDI categories.

Re: Thank you for your helpful suggestion. To improve interpretability, we have added visual annotations to Figure 6 to indicate key divergence points in the association between SDI and age-standardized IHD burden.

Revised figure legend (Figure 6):

“Fig 6 SDI-based age-standardized rates of DALYs and ASDR for IHD attributable to HSBP. (A) SDI-based age-standardized rates of DALYs in 21 regions from1990 to 2021. (B) SDI-based ASDR in 21 regions from1990 to 2021. (C) SDI-based age-standardized DALYs across countries in 2021. (D) SDI-based ASDR across countries in 2021. Arrows point to major inflection points (SDI ≈ 0.7) where IHD burden begins to decline with increasing SDI. SDI: Socio-demographic Index; DALYs: Disability-adjusted life years; ASR: Age-standardized rate.”

Discussion:

The extended analysis of sex- and region-specific disparities is stronger. Emphasize how findings support global and local initiatives for hypertension control.

Re: We appreciate your thoughtful suggestion. In response, we have expanded the Clinical Significance section to highlight the practical implications of our findings for global and region-specific hypertension control efforts.

Revised Limitations and Clinical Significance: “Despite these limitations, our analysis provides meaningful insights into the demographic and geographic variability of IHD burden attributable to hypertension. By identifying high-risk populations and regions, the findings offer a basis for targeted interventions and policy development. The observed parabolic relationship between SDI and disease burden suggests that uniform strategies may be suboptimal. In lower-SDI settings, efforts should focus on early detection, treatment accessibility, and primary care integration. In higher-SDI contexts, optimizing adherence, promoting healthy lifestyles, and addressing care gaps remain key. Aligning hypertension control strategies with health system capacity and socioeconomic conditions may help reduce global disparities in cardiovascular outcomes.” (Lines 468-477).

Limitations

Acknowledge the inability to infer causality from ecological-level GBD data.

Re: Thank you for this valuable suggestion. As suggested, we have clarified in the Limitations section that the ecological design of the GBD data.

“Moreover, the ecological nature of GBD estimates—derived from population-level rather than individual-level data—further limits the ability to draw causal inferences across countries or subpopulations.” (lines 452-454).

Note that despite robust modeling, within-country inequalities in SDI and BP control are not captured

Re: Thank you for this insightful comment. We have revised the Limitations section to emphasize that national-level indicators such as SDI and blood pressure may obscure important within-country disparities.

“Finally, the use of national-level aggregates—such as SDI and blood pressure data—may obscure substantial within-country disparities, particularly in large or socioeconomically diverse countries. For example, individuals in low-SDI countries may, in some contexts, have access to high-quality healthcare, while marginalized groups in high-SDI countries may face significant barriers to hypertension diagnosis and treatment. Future research should incorporate individual-level and longitudinal data to improve risk estimation and better elucidate the complex interactions among HSBP, socioeconomic status, and IHD outcomes.” (Lines457-464).

Conclusion:

Now appropriately focused and policy-oriented.

Re: Thank you for the positive feedback.

---

## [Editor Report · Decision Letter 2]

21 Apr 2025

Global trends and disparities in ischemic heart disease attributable to high systolic blood pressure, 1990–2021: insights from the Global Burden of Disease study

PONE-D-25-01876R2

Dear Dr. Guo,

We’re pleased to inform you that your manuscript has been judged scientifically suitable for publication and will be formally accepted for publication once it meets all outstanding technical requirements.

Kind regards,

Thien Tan Tri Tai Truyen, M.D.

Academic Editor

PLOS ONE
---

## [Editor Report · Acceptance letter]

PONE-D-25-01876R2

PLOS ONE

Dear Dr. Guo,

I'm pleased to inform you that your manuscript has been deemed suitable for publication in PLOS ONE. Congratulations! Your manuscript is now being handed over to our production team.

Kind regards,

on behalf of

Dr. Thien Tan Tri Tai Truyen

Academic Editor

PLOS ONE